# Uptake of Oral HIV Pre-Exposure Prophylaxis (PrEP) and Associated Factors among Female Sex Workers in Tanga, Tanzania

**DOI:** 10.3390/v15102125

**Published:** 2023-10-20

**Authors:** Veronica O. Martin, Novatus A. Tesha, Bruno F. Sunguya

**Affiliations:** School of Public Health and Social Sciences, Muhimbili University of Health and Allied Sciences, United Nations Rd., Dar Es Salaam 11103, Tanzania; sunguya@muhas.ac.tz

**Keywords:** female sex workers, pre-exposure prophylaxis, HIV, key population, oral PrEP

## Abstract

Pre-exposure prophylaxis (PrEP) prevents HIV infection among female sex workers (FSW). WHO recommends the use of Tenofovir disoproxil fumarate for use in oral PrEP regimens (TDF). Emtricitabine (FTC) 200 mg/Tenofovir Disoproxil Fumarate (TDF) 300 mg (Truvada) daily is the approved PrEP regimen in Tanzania. Evidence is limited on oral PrEP uptake and its associated factors in countries with a high burden of HIV, such as Tanzania. This study aimed to examine the uptake of oral PrEP and its associated factors among FSW in the Tanga region of Tanzania. This community-based cross-sectional study was conducted among 428 FSW. Data were collected through face-to-face interviews and analysed using STATA version 17 and RDSAT. Logistic regression was used to examine the associations of independent factors and PrEP uptake among study participants. About 55% of the recruited FSW used oral PrEP. FSW with three or more children were 2.41 times more likely to take oral PrEP (AOR 2.41, 95% CI: 1.08–4.25, *p* < 0.05). Moreover, those with a positive attitude were more likely to use oral PrEP (AOR 2.8, 95% CI: 1.88–4.17, *p* < 0.05). Poor belief was a barrier to PrEP use, and side effects of the drugs were a reason for the discontinuation of PrEP services. Most of the participants preferred PrEP services to be provided in the community. Oral PrEP uptake was 55%. Efforts to scale up PrEP for FSW should address misconceptions regarding PrEP, PrEP sensitization, and improving access through community-based intervention.

## 1. Introduction

Oral PrEP is the daily use of antiretroviral (ARV) medications by HIV-uninfected sexually active persons to prevent HIV infection before exposure to the virus. Tenofovir disoproxil fumarate is recommended by the WHO for use in oral PrEP regimens (TDF) [1]. In the Tanzania context, the current guidelines for the management of HIV and AIDS recommend the daily use of Emtricitabine (FTC) 200 mg/Tenofovir Disoproxil Fumarate (TDF) 300 mg (Truvada) PO [2]. The use of PrEP has become a female-controlled HIV prevention method among FSW who are unable to negotiate condom use during sexual activity [3].

Female sex workers (FSW) are among the vulnerable and key population bearing the higher brunt of HIV infection as compared to the general population, particularly young females aged between 15 and 24 years [4]. The group is vulnerable as it has been underserved by HIV prevention strategies [5]. The introduction of oral pre-exposure prophylaxis (PrEP) to protect against HIV infection has proven itself to be a safe and effective HIV infection prevention method for FSW among high-risk populations.

Although the burden of HIV/AIDS has dropped in the general population globally, it remains high and persistent among FSW despite a variety of preventive interventions [6]. This achievement has been slowed down by drug resistance, including the new regimes of ART [7]. Approximately 1 in 1000 PrEP users in clinical trials experience medication resistance, which is a modest risk. Nearly all of those who developed drug resistance when taking PrEP already had an acute, undiagnosed HIV infection. To prevent drug resistance, HIV testing must be performed prior to the commencement of PrEP. As a result, PrEP is anticipated to lessen the public health burden of HIV medication resistance by reducing the number of new HIV infections that would otherwise require lifelong therapy and carry a high risk of developing drug resistance [6]. Evidence suggests that the key population contributes to more than half of all people living with HIV globally, and they account for about 39% of all new HIV infections in sub-Saharan countries [8]. In Tanzania, the HIV prevalence has also dropped from 7% in 2003 to 4.6% in 2018, but it is twice as high among women compared to men. FSWs contribute 14–35% to HIV prevalence among the key population [9,10].

The HIV/AIDS prevention interventions program that mostly targeted key populations showed that cash transfers for safer sex, counselling to promote condom use, and oral PrEP services are more effective and ethical options among FSWs [11]. The Determined, Resilient, Empowered, AIDS-free, Mentored, and Safe women (DREAMS) program was designed mainly to reduce the burden of HIV and early pregnancy. The oral PrEP intervention is one of the primary interventions of the DREAMS program, and it reports that the uptake of oral PrEP among FSW in Sub-Saharan Africa has been very low. Some of the barriers reported include inconsistent access to oral PrEP and the incompetency of a nonprofit organization that serves the community to provide oral PrEP services to the population [3].

In achieving UNAIDS’s ambitious target of “95-95-95” by 2025, in which the main aim is to have 95% of people living with HIV (PLHIV) know their HIV status, 95% of diagnosed cases be on ART, and 95% attain an undetectable viral load, the combination prevention strategy was introduced. This strategy involves the integration of condom promotion and community empowerment incorporated with biomedical interventions such as oral PrEP access and its use to reduce the national HIV incidence by 90% by 2030. In the area where oral PrEP intervention has been introduced, it shows a good integration with other HIV prevention programs among FSW [2].

Initiation of oral PrEP intervention among sex workers is highly promising in reducing HIV infections, as oral PrEP reduces the need to consent to condom use and avoids inconsistent condom use during sexual acts [5]. Inconsistent use of condoms among FSWs may be due to limited urgency, condom breakage or slippage, sexual or physical violence, client refusal or coercion, or clients who pay more but for not using a condom [11,12]. The interest level in oral PrEP and willingness to use oral PrEP among high-risk groups, including FSW, is high but does not guarantee uptake of oral PrEP services [13,14]. In Tanga, evidence is lacking on oral PrEP access and uptake among FSW in the prevention of HIV infection. This study therefore intended to assess the uptake of oral PrEP and associated factors among female sex workers in the Tanga region using Emtricitabine (FTC) 200 mg/Tenofovir Disoproxil Fumarate (TDF) 300 mg (Truvada) daily. The study offers the first evidence on the use of PrEP and its determinants thereof in Tanzania and in the region with potential high transmission from key and vulnerable populations in a context of generalized epidemic.

## 2. Materials and Methods

### 2.1. Study Design and Setting

This community-based analytical cross-sectional study was conducted among female sex workers in the Tanga region of Tanzania in August 2022. The region is made up of nine councils, which are Korogwe, Handeni, Mkinga, Muheza, Pangani, Tanga, Lushoto, Bumbuli, and Kilindi. The region is famous for its robust economy from agriculture and natural resource-based activities like forestry, fishing, and mining. It borders Kenya, offering significant economic and business exchanges and attracting people to the region. The region has an HIV prevalence of 5% [15], slightly higher than the national average. Five districts, which are Korogwe, Muheza, Handeni, Tanga, and Lushoto, were selected for this study because of their higher HIV prevalence and the large number of business centres that are commonly attractive for female sex workers (FSWs) [1].

### 2.2. Study Population

The selected participants included women aged above 18 who exchanged sex for money or goods in the Tanga region. The study excluded FSWs who self-reported to be HIV positive and on ART.

### 2.3. Sample Size

The sample size was determined using a single population proportion formula: S = Z^2^ × P × (1 − P)/E2. The proportion of 50% for the infinite population was used because the FSW population is hard to reach. In cases like this, a sampling frame cannot be generated [16]. The standard normal deviation of 1.96 using the 95% confidence interval (Z) and 5% margin of error (E) was used, giving a minimum sample size of 386. A 10% non-response was considered, and therefore the minimum sample size was 428.

### 2.4. Sampling Procedures

Respondent-driven sampling (RDS) was used to facilitate the recruitment of study participants for which a sampling frame was not available. Initially, peer educators were recruited as seeds, who then suggested other individuals of the same demographic characteristics to the research team for involvement in a series of waves until the necessary sample size was achieved. A minimum of four and a maximum of seven seeds from each council (selected 5 councils) were recruited at the beginning of the study, resulting in a total of 25 seeds. Four seeds (16.0%) were in the age category of less than 25 years, 13 (52%) were between the ages of 26 and 35 years, and 8 (32.0%) were above the age of 35 years. In this study, peers were used to recruit other peers to ensure the privacy and anonymity of the study participants. Hence, a total of 6 waves of recruitment were achieved, with the most seeds recruiting 3 to 4 participants in the study.

### 2.5. Variables and Their Measurements

The outcome variable was the uptake of oral PrEP. This was estimated by taking the number of FSW who ever used oral PrEP (numerator) and dividing it by all FSW enrolled in the study (denominator). Independent variables, including demographic and socioeconomic characteristics (age, sex, education level, marital status, number of children, household characteristics and possessions, household food insecurity, social support, and employment status), were adopted from the questionnaire that was employed in the Demographic Health Survey [17]. Behavioural factors like sexual history and risk factors, substance and drug use [18], and belief and attitude were also independent variables. For age, the year of birth that reflected a number of years was collected; for reasons of data protection, the day of birth and month were not collected [17]. For their current marital status, participants were asked if they were married or living together, divorced/separated, widowed, or never married or lived together [17]. For the number of children, respondents were asked to mention the number of live children they have. A number of children were grouped for those who have one or two children, 3–5 children, and more than six children. Education level was categorized into primary school, post-training primary school, ordinary level secondary, advanced level secondary (high school), and college or university [17]. For the employment status, respondents were required to respond on their employment status as not employed and employed (the employed group was then categorized into temporary employed and permanent employed). The tool was adopted from the Tanzania Demographic Health Survey (TDHS) and malaria indicators survey [17].

Individual and household food insecurity was assessed using the HFIAS scale [19]. This scale is used to measure household food security using a 4-week (30-day) recall. The assessment of the household food insecurity access scale (HFIAS) was performed using a nine-item questionnaire by the Food and Nutrition Technical Assistance Project (FANTA), with a minimum score of 0 and a maximum score of 27. The frequency of occurrence during the past 30 days was summed, and the average of household food insecurity was calculated by dividing the sum of HFIAS scores in the sample by the number of HFIAS scores in the sample. The higher the score, the more food insecurity the household experiences. The lower the score, the less food insecurity the household experiences. Food insecurity was categorized into 4 groups (after responding to nine questions) according to HFIAS categories: severely insecure, mildly insecure, moderately insecure, and secure.

Economic status was examined using the weighted wealth index scale. The wealth index was used as a measure of household economic status by considering household asset ownership (such as televisions and bicycles) and living conditions. The wealth index was computed using principal component analysis and factor analyses of household asset ownership. The factor loadings, which are sample weights, are summed to generate the weighted wealth index, which was divided into quantiles into poorest, poorer, middle, richer, and richest wealth quantiles [20].

Social support was measured using the Multidimensional Scale of Perceived Social Support (MSPSS) with a scale of 1 to 5 and eight questions. The total scores of all eight questions were summed and divided by 8 in order to obtain the mean score, in which the higher the mean score, the greater the family’s socio support [21]. For behavioural factors, the adapted structured questionnaire from the study conducted in Dar es Salaam contains questions that explore the behavioural-related factors associated with uptake of oral PrEP, like sexual history and risk factors, attitude towards oral PrEP use, as well as substance and drug use [18].

Sexual risk behaviour was examined with the administered questionnaire used to assess the perceived risk of getting HIV infection [22]. The term sexual risk behaviour was used to help concentrate on behaviour rather than a person; this was measured by asking about the number of sexual partners. It was categorized as those with more than 3 sexual partners coded as at risk and less than 2 with no risk. Consistency of condom use was measured using the Likert scale, with scores ranging from 1 (always), 2 (most of the time), 3 (sometimes), and 4 (never). Those who responded with answer 1 (always) were coded as having no risk, while those who answered with 2, 3, or 4 were coded as at risk. Participants who responded that they had sex with HIV-positive individuals (yes or no) were treated as being at risk.

Substance and drug abuse were examined using the tools for substance and drug abuse adopted from a study conducted in Dar es Salaam on flash blood use among women [23]. The respondents were asked if they had used alcohol or substances in the past 3 months. The response was categorized as either yes or no, and if yes, a quantification of the number of bottles of beer consumed in a typical day was recorded. If the participant drinks seven to nine beers on one occasion, it is classified as risk drinking, and if he/she drinks 1 to 2 days in a week and drinks only 1 to 2 beers, it is classified as non-risk.

Self-efficacy was assessed using the Strengths Self-Efficacy Scale (SSES), which involved assessing study participants self-belief in their ability to create a sense of personal strength [24]. The self-efficacy was measured using a Likert scale on how much study participants agreed or did not agree on eight statements. The final score was obtained by taking the mean of all responses, and the high scores indicated greater self-efficacy among study participants.

Medical contraindications were assessed using an adopted questionnaire for the presence of medical conditions among study participants that impacted the uptake of oral PrEP, as explained in the PrEP guidelines by the government of Barbados [25]. This was measured by asking and assessing the presence of any medical contraindications, especially diseases such as renal diseases, liver failure, and drug sensitivity.

A questionnaire on services delivery points was used to assess preferences for service delivery points [26]. Participants were asked to choose between 2 delivery points for PrEP services that would be convenient for them, either at community- or hospital-based delivery points for PrEP services, and their impact on oral PrEP uptake among study participants.

Belief and attitude about oral PrEP use were measured by using 6 adopted and modified Likert scale questions from the attitude and belief assessment by the Centres for Disease Control (CDC) [27], on a scale of 1 to 5, from strongly disagree to strongly agree. The score ranges between 4 and 5 were regarded as good, while those with less than 3 were regarded as poor.

### 2.6. Data Collection Tools and Procedures

The standardized questionnaire was administered to 428 FSWs in Tanga. Peer educators pre-tested the tools to guarantee cultural and peer relevance. The questionnaire was prepared in English, translated into Swahili, and administered by trained research assistants and the principal investigator for data collection. All interviews were performed in Swahili. The recruitment process began with five [5] seeds representing various FSW groups in terms of age, residence, and type of sex worker (street, brothel, club, and bar-based). Peer health educators were the first point of contact with the seeds. Data were gathered utilizing a pre-tested Swahili questionnaire among FSW in either hotspots, houses, or ghettos. Four research assistants were hired and trained on the study protocol, questionnaire, ethics in research, and insisting that working with FSW requires compassion. All study participants signed an informed consent to participate, and face-to-face interviews were performed in private rooms at their catchment areas to ensure confidentiality. Data were collected depending on the availability of study participants. About 15 to 30 min were used per participant. Each participant received invitation coupons at the end of the interview to recruit more participants to join the study.

### 2.7. Data Management and Analysis

Analyses were conducted using both descriptive and logistic regression methods. Descriptive statistics were used to estimate the proportion of FSW with oral PrEP uptake by taking the number of FSW who ever used oral PrEP (numerator) divided by all FSW enrolled in this study (denominator) and assessing the characteristics of participants. The Respondent Driven Sampling Analysis Tool (RDSAT) was used to estimate an asymptomatic unbiased estimator of population parameters. Therefore, an adjusted proportion with a 95% confidence interval (CI) for FSW on the level of uptake of oral PrEP was calculated and presented in percentage.

Both bivariate and multiple logistic regression analyses were used to determine the factors associated with oral PrEP uptake. Bivariate logistic regression was performed to calculate the crude odds ratio (COR). All variables with *p* < 0.2 were included in multiple logistic regression to calculate the adjusted odds ratio (AOR).

A Likert scale with five levels from strongly disagree to strongly agree ranging from 1 through 5, respectively, was used to examine belief and attitude towards oral PrEP uptake. Individual response scores were computed. The score above neutral/3 was defined as positive attitudes/beliefs; otherwise, it was poor. The findings were summarized by proportion and frequency, and bivariate and multiple logistic regression analyses were conducted to determine the association between PrEP uptake and belief in and attitude towards it.

## 3. Results

### 3.1. Characteristics of Study Participants

The mean age of the 428 FSWs recruited was 31.8 (SD ± 7.3) years, and more than three-quarters (76.4%) were aged above 25 years. The majority (79.4%) had secondary or above-level education, and less than 30% reported having 3 or more children. The majority (82.9%) were not in a union (divorced/separated, widowed, never married, and/or not living with someone), 49.3% had no employment, and about two-thirds reported having food security but experienced poor family/social support and had low self-efficacy (Table 1).

The majority (83.4%) had more than three sexual partners in the previous month. In the most recent sexual interaction, more than two-thirds (67.5%) reported to have used condoms, although among those who reported to have used condoms in the most recent sexual interaction, more than 80% claimed to use them infrequently or never. About 86% of the participants used alcohol, while 6.3% reported using injectable narcotics (Table 2).

More than three-quarters (79.7%) of the participants reported being willing to take PrEP. More than 90% of the FSWs preferred PrEP services to be provided in the community instead of the health facility (Figure 1).

### 3.2. Pre-Exposure Prophylaxis (PrEP) Uptake

Among the 428 FSWs recruited in this study, 54.7% had ever used pre-exposure prophylaxis (PrEP). Of them, more than 97% reported using PrEP currently. Among FSWs who stopped using PrEP, 80% reported that the reason was PrEP side effects. The reported side effects include headaches, nausea, and weight loss.

### 3.3. Factors Influencing Pre-Exposure Prophylaxis (PrEP) Use among FSW

In multivariate analysis, the number of children and FSW attitude were significantly associated with PrEP uptake. FSWs with three children were 2.14 times more likely to use PrEP compared to those with no children. Moreover, FSWs with a positive attitude had a 2.83 times higher risk of PrEP uptake as compared to those with a negative attitude (Table 3).

### 3.4. Belief and Attitude towards oral PrEP Uptake

More than half (54.4%) of the FSWs in Tanga had good beliefs towards PrEP uptake; however, only 45.6% had a positive attitude towards PrEP uptake (Table 4).

## 4. Discussion

This cross-sectional study used the respondent-driven sampling method to collect data among 428 FSWs in the Tanga region. Only 54.7% of the recruited FSWs had ever used PrEP, and among them, 97% were active users at the time of the study. The number of children and attitude towards PrEP were independent predictors of PrEP use. Risky sexual behaviours did not predict PrEP uptake in this population.

In this study, slightly more than half (54.7%) of FSWs had ever used PrEP. This is higher compared to 5.3% observed in 2021 among key populations in Tanzania and 8% in Dar es Salaam; however, it remains lower compared to the UNAIDS’ global target of reaching 90% of key populations [5,26,27]. Although the magnitude of FSWs utilizing this evidence-proven effective intervention for HIV prevention has grown significantly over the two-year period, it still highlights the need for more work to reach the globally ratified national target for this key population. There is still a lack of thorough knowledge on PrEP among FSW. The reduction of new HIV infections among FSW and the key population at large will be hindered by such low PrEP coverage [5].

The observed increase in PrEP use among FSWs in the Tanga region may be an indication of the responsive nature of HIV prevention interventions carried out in the country (5). PrEP uptake among FSW observed in this study is higher than that observed in Kenya (44%) and South Africa (16%) [2,6]. On the other hand, PrEP uptake was 82.4% with 73.4% retention after 12 months in Dakar, Senegal. These differences highlight the context-specific approaches needed when designing and carrying out PrEP-related interventions. Moreover, in this study, only 2% of FSWs that initiated PrEP discontinued; this is contrary to Kenya, South Africa, and the United States, where up to 50% or more clients discontinued within the first one to six months of PrEP use [2,9]. This calls for more effort to be directed toward initiating FSWs on PrEP in Tanzania and the continuation of current retention strategies by tracking clients who withdrew from PrEP services.

Addressing PrEP non-use among FSWs in the region called for efforts to understand its determinants. This study found two independent factors associated with the use of PrEP. The number of children that the FSW has was an independent predictor of oral PrEP uptake. That is, the higher the parity, the higher the odds of PrEP usage. This may indicate the feelings of FSWs in protecting their children through self-prevention of HIV themselves. It may also signify awareness of risky behaviours that older women with a higher number of children have compared to those with none or fewer children. The evidence may also indicate that FSW do not wish their children to lack basic needs and become orphans after contracting HIV and dying. This single piece of evidence may be useful in planning interventions among key vulnerable populations (KVPs). Targeting FSWs with no or fewer children may help reduce the burden of new infections, but also using the multiparous FSWs as mentors and peer educators may be a useful approach to increasing the proportion of FSWs who access and use PrEP. However, a study performed in Sub-Saharan Africa among adolescent girls and young women to assess factors associated with PrEP use showed no association between the number of children and PrEP use [28,29,30,31].

FSWs with a positive attitude have higher odds of PrEP usage. Moreover, expectations of receiving more positive feedback from others and confidence in their ability to use PrEP had a higher odds ratio for PrEP uptake. This complements the qualitative evidence that women who learn the benefits of PrEP have favourable views towards PrEP uptake [32].

Lack of proper education on PrEP for the prevention of HIV may have determined poor beliefs and attitudes towards PrEP use among FSW. Poor belief that PrEP could lower the chances of acquiring a new HIV infection, daily use may interfere with work, and oral PrEP being linked with being HIV positive led to low uptake of oral PrEP among FSW. Overall, 45.5% of FSWs had poor beliefs towards PrEP uptake. The belief that PrEP use could lower their chances of acquiring HIV infection was present in only 44.4% of the participants. These findings could complement the findings from South Africa, where it was reported that the common barriers to PrEP use were due to limited knowledge about PrEP, which led to doubts regarding its efficacy, perceptions of HIV risk, HIV stigma associated with PrEP use, as well as concerns pertaining to its side effects and medication interactions [1]. FSW who were aware of the advantages of PrEP were more likely to use it as an HIV prevention strategy, which helped increase the uptake of PrEP. This is consistent with the existing literature assessing the importance of PrEP use and disclosure to parents of adolescent girls in Kenya [3].

The evidence presented in this study should be discussed in light of a few limitations. First, this being a cross-sectional study, it may not give a causal association rather than just associations. However, it conforms with evidence presented elsewhere and further adds to the literature on effective interventions for HIV prevention among key and vulnerable populations. Moreover, the study was conducted in just one region with the highest average burden of HIV in the country. Nevertheless, the evidence presented here is unique but relates to other key populations in the country and regions with similar context. Data on PrEP use were self-reported and therefore subject to reporting bias owing to the sensitivity of the intervention. Participants were recruited using a respondent-driven sampling procedure. This could have led to the overrepresentation of participants with similar characteristics. However, this is the best recruitment process for such a hidden population owing to likely discrimination due to customs and traditions. The duration of oral PrEP or adherence data for PrEP use could not be ascertained in this study. High PrEP discontinuation is reported at 1 month [11]. Despite these limitations, our study offers the first evidence on the use of PrEP and determinants thereof in Tanzania and in the region with potential high transmission from key and vulnerable populations in a context of generalized epidemic.

## 5. Conclusions

PrEP uptake among FSW was still low in Tanga, despite widespread PrEP rollout in Tanzania. Our evidence found only 54.7% of FSWs were using PrEP, which is widespread and free for key and vulnerable populations. This signifies a need to increase efforts towards the use of this key and vulnerable population. Having three or more children and a positive attitude towards oral PrEP influenced PrEP uptake among FSWs. Poor belief was reported to be a common barrier to PrEP use among FSW. Side effects of the drugs were reported to be the reason for the discontinuation of PrEP services. This suggests the need for routine screening for PrEP use among FSW and the development of targeted intervention to mitigate the issue of discontinuation and poor belief and attitude towards PrEP uptake. Most of the participants preferred PrEP services to be provided in community settings rather than in health facility settings.

## Figures and Tables

**Figure 1 viruses-15-02125-f001:**
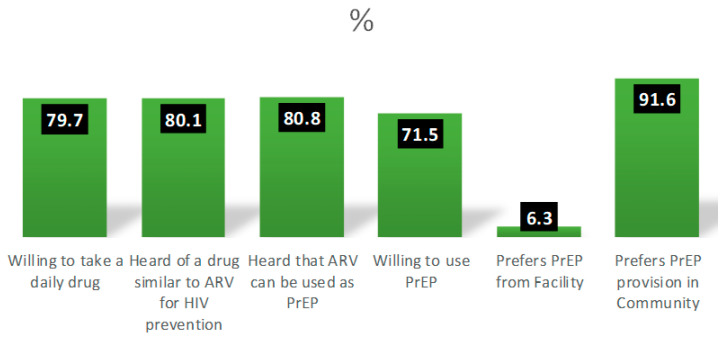
Readiness to use pre-exposure prophylaxis (PrEP) among FSW (n = 428).

**Table 1 viruses-15-02125-t001:** Characteristics of participants in the Tanga region (n = 428).

Characteristic	n	%
Age Mean (SD) 31.8 (±7.3)		
18–25	101	23.6
>25	327	76.4
Education level		
No formal education	14	3.3
Primary education	74	17.3
Secondary education or above	340	79.4
Marital status		
Married/Living together Divorced Widow	73 120 39	17.1 28 9.1
Never married	196	45.8
Number of children		
No children	129	30.1
1 to 2	181	42.3
≥3	118	27.6
Employment		
Not employed	211	49.3
Temporary employed	170	39.7
Permanent/stable employment	47	11.0
Household food insecurity		
Severely Insecure	127	29.7
Mild insecure	18	4.2
Moderate insecure	22	5.1
Secure	261	61.0
Weighted wealth index		
Poorest Poorer	110 134	25.7 31.3
Middle	119	27.8
Richer Richest	51 14	11.9 3.3
Social support		
Low social support	288	67.3
High/good social support	140	32.7
Self-efficacy		
Low self-efficacy	279	65.2
Greater/sufficient self-efficacy	149	34.8

**Table 2 viruses-15-02125-t002:** Sexual risk behaviours among participants (n = 428).

Variable	n	%
Number of sex partners in the last month		
Low risk (<3)	71	16.6
High risk (3 or more)	357	83.4
Used a condom in the last sexual encounter		
Yes	139	32.5
No	289	67.5
Consistency of condom use		
Never	68	15.9
Sometimes	287	67.1
Always	73	17.1
Able to obtain a male condom when in need (n = 364)		
Yes	128	35.2
No	236	64.8
Alcohol use in the past 3 months		
Yes	367	85.7
No	61	14.3
Injected drugs for fun or relaxation in the past 3 months		
Yes	27	6.3
No	401	93.7

**Table 3 viruses-15-02125-t003:** Bivariate and multivariate logistic regression for factors associated with oral PrEP uptake (n = 428).

Variable	n	PrEP Uptake	Bivariate Log Reg		Multivariate Log Reg	
		n (%)	COR (95% CI)	*p*-Value	AOR (95% CI)	*p*-Value
Age						
≤25	101	54 (53.5)	1			
>25	327	180 (55.0)	1.07 (0.68–1.67)	0.78		
**Education level**						
No formal education	14	12 (85.7)	1		1	
Primary education	74	40 (54.1)	0.25 (0.04–0.94)	0.04	0.35 (0.07–1.88)	0.22
Secondary education/above	340	182 (53.5)	0.19 (0.04–0.87)	0.03	0.29 (0.06–1.44)	0.13
**Marital status**						
Married/Living together	73	45 (61.6)	1		1	
Divorced	120	58 (48.3)	0.58 (0.32–1.05)	0.07	0.62 (0.31–1.24)	0.17
Widow	39	26 (66.7)	1.24 (0.55–2.81)	0.6	1.34 (0.52–3.45)	0.55
Never married	196	105 (53.6)	0.72 (0.42–1.24)	0.24	1.12 (0.56–2.24)	0.76
**Number of children**						
No children	129	72 (55.8)	1		1	
1 to 2	181	86 (47.5)	0.72 (0.46–1.13)	0.15	0.81 (0.46–1.45)	0.48
≥3	118	76 (64.4)	1.43 (0.86–2.39)	0.17	2.14 (1.08–4.25)	0.03 *
**Employment**						
Not employed	211	114 (54.0)	1		1	
Temporary employed	170	86 (50.6)	0.87 (0.58–1.31)	0.50	0.92 (0.57–1.49)	0.72
Permanent/stable employment	47	34 (72.3)	2.23 (1.11–4.45)	0.02	2.11 (0.95–4.7)	0.07
**Household food insecurity**						
Severely insecure	127	65 (51.2)	1			
Mild insecure	18	10 (55.6)	1.2 (0.44–3.22)	0.73		
Moderate insecure	22	17 (77.3)	3.24 (1.13–9.32)	0.03		
Secure	261	142 (54.4)	1.04 (0.7–1.57)	0.55		
**Wealth index**						
Poorest (Low)	110	60 (54.5)	1			
Poorer	134	75 (56.0)	1.06 (0.64–1.76)	0.82		
Middle	119	60 (50.4)	0.85 (0.5–1.43)	0.53		
Richer	51	28 (54.9)	1.01 (0.52–1.98)	0.97		
Richest	14	11 (78.6)	3.06 (0.81–11.56)	0.10		
**Number of sex partners in the last month**						
Low risk (<3)	71	36 (50.7)	1			
High risk (3 or more)	357	198 (55.5)	1.21 (0.73–2.02)	0.46		
**Used a condom in the last sexual** **encounter**						
Yes	139	71 (51.1)	1			
No	289	163 (56.4)	1.24 (0.83–1.86)	0.30		
**Belief**						
Poor	195	92 (47.2)	1		1	
Good	233	142 (60.9)	1.75 (1.19–2.57)	0.01	1.51 (0.94–2.43)	0.09
**Attitude**						
Negative	233	101 (43.3)	1		1	
Positive	195	133 (68.2)	2.8 (1.88–4.17)	0	2.83 (1.75–4.57)	0 **

AOR—Adjusted odds ratio; *p* * < 0.05, and *p* ** < 0.01

**Table 4 viruses-15-02125-t004:** Beliefs and attitudes toward PrEP uptake among FSW in Tanga (n= 428).

Variable	n	%
**Belief**		
Poor	195	45.6
Good	233	54.4
**Attitude**		
Negative	233	54.4
Positive	195	45.6

## Data Availability

The data are available upon request.

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
