# Peer review of "Uptake of Oral HIV Pre-Exposure Prophylaxis (PrEP) and Associated Factors among Female Sex Workers in Tanga, Tanzania"

_viruses, 2023, doi:10.3390/v15102125_

Round 1
Reviewer 1 Report
The Manuscript ID: viruses-2550054, entitled: “Uptake of Oral HIV Pre-Exposure Prophylaxis (PrEP) and Associated Factors among Female Sex Workers in Tanga, Tanzania” brings relevant information about PrEP among Female Sex Workers”. This paper will bring a relevant contribution and considerations to researchers working with PrEP, and HIV treatment and deserves to be published. However, the present manuscript requires some amendments before its publication.
Abstract: Should include the medication name, dose and frequency of use for PrEP. It is included only in the introduction.
Introduction:
1) Why did the authors choose Emtricitabine (FTC) 200 mg/Tenofovir Disoproxil Fumarate (TDF) 300 mg (Truvada) daily? Is it the only oral PrEP available in Tanzania? Or in the world? Please, justify.
2) Include on page 2, line 72: a paragraph including the main objective of the study, with the drug used, concentration and dose frequency. “This study, therefore, intended to assess the uptake of oral PrEP and associated factors among female sex workers in the Tanga region using Emtricitabine (FTC) 200 mg/Tenofovir Disoproxil Fumarate (TDF) 300 mg (Truvada) daily”
Material and Methods:
1) Please correct the grammar on Page 2, line 92: “The proportion of 50% for the infinite population was used as the FSW population is hard to reach population in which sampling frame cannot be easily generated (15).”
Not many mistakes detected
Author Response
We are thankful for the comments. We have addressed them. Find the attached.

Reviewer 2 Report
In this highly relevant work, the authors examine the uptake of oral PrEP and its associated factors among female sex workers (FSW) in Tanga region in Tanzania. While the study is comprehensive and highly relevant, the manuscript could benefit from a clearer exposition of what the main conclusions and associations are, why they are statistically significant, and what advancement is being made through the work, that is not available from the prior literature in HIV. Overall, I believe that the manuscript merits publication in Viruses, but the authors should address the following comments.
Major Comments:
The authors should clearly highlight the novelty of the work as to why the results are significant, and what new advancements and biological insights are being made in understanding and management of HIV infection.
[It is mentioned towards the end of the Discussions and Conclusions sections, but would benefit from clearer exposition in the Introduction perhaps.]
1.. Lines 41-43: “Evidence suggests that the key population contributes to more than half of all People living with HIV globally and they account for about 39% of all new HIV infections in Sub-Saharan countries”. The authors should include some commentary on the effect on drug resistance, whether transmitted or acquired, which could be a serious risk factor, in general (https://www.who.int/news-room/fact-sheets/detail/hiv-drug-resistance). Drug resistance continues to be a persistent problem affecting the lives of millions of patients worldwide [Biswas, Avik, et al., Elife 8 (2019), Biswas, A. et al., Plos one, 17(1), e0262314, Choudhuri, Indrani, et al.,The Journal of Physical Chemistry B 126.50 (2022), Charpentier, Charlotte, et al. "2022 Update of the Drug Resistance Mutations in HIV-1."], including against the latest generation of ART drugs [Li, Min, et al., Science Advances 9.29 (2023): eadg5953.] with reports of emerging pan-resistant HIV [Puertas, M. C. et al. Lancet Microbe (2020)]. To make the manuscript text more complete, the authors should briefly comment on drug resistance and include the appropriate citations, as above.
2. Lines 250-251: Among FSWs 250 who stopped using PrEP, 80% reported a reason being PrEP side effects. The authors should elaborate on what the side effects are (given that a majority reason for PreP stoppage is side effects) and if any associations were performed with particular detrimental side effects.
3. Lines 296-297: While discussing the number of children as a deterministic factor for PrEP usage, the authors impute, “It may also signify awareness of risky behaviors that older women with higher number of children have compared to those with none or fewer children.” The authors should clearly justify such such a correspondence, causal relationship or statistical significance of it based on the methodology used.
This argument also forms the basis of some recommendations to reduce infection that the authors suggest.
Minor Comments:
1. Acronyms such as PLHIV (line 57) should be dfined on first usage.
2. Line 40: Usage of “key population”. The authors should clarify that key population refers to FSW.
Author Response
Thanks very much for the review and comments. We have addressed the comments and our responses are the the attachment.

Reviewer 3 Report
An excellent study; only a few minor suggestions:
Line 29-31: Incomprehensible; please rephrase the sentence.
Line 67: …..due to limited agency…..?
Line 133: FANTANA?
Line 160: Likert, and not likert
Line 163: ……were at treated…….
Line 172: …..to creates…
Line 174: …..does not agreed…
Line 211: RDS?
Line 217: ….crudes odds ratio…..
Excellent; only a few grammatical errors/typos.
Author Response
We are very thankful for the comments. We have reacted on then on the attachment with this note.
